# Predicting the Stability of Organic Matter Originating from Different Waste Treatment Procedures

**DOI:** 10.3390/ijerph20032151

**Published:** 2023-01-25

**Authors:** Yan Wang, Lekun Tan, Patricia Garnier, Sabine Houot, Julie Jimenez, Dominique Patureau, Yang Zeng

**Affiliations:** 1Sino-French Research Institute for Ecology and Environment (ISFREE), School of Environmental Science and Engineering, Shandong University, 72 Bing Hai Avenue, Qingdao 266237, China; 2Qingdao ProBio Biotech Co., Ltd., Block B, Building 3, Haichuang Center, Blue Silicon Valley, Qingdao 266200, China; 3AgroParisTech, INRAE, Université Paris Saclay, UMR ECOSYS, F-78850 Thiverval Grignon, France; 4French National Institute for Agriculture, Food, and Environment, University of Montpellier, LBE, INRAE, 102 Avenue des Etangs, F-11100 Narbonne, France

**Keywords:** anaerobic digestion, composting, soil, organic matter, stability, soil application

## Abstract

Recycling organic wastes into farmland faces a double challenge: increasing the carbon storage of soil while mitigating CO_2_ emission from soil. Predicting the stability of organic matter (OM) in wastes and treatment products can be helpful in dealing with this contradiction. This work proposed a modeling approach integrating an OM characterization protocol into partial least squares (PLS) regression. A total of 31 organic wastes, and their products issued from anaerobic digestion, composting, and digestion-composting treatment were characterized using sequential extraction and three-dimension (3D) fluorescence spectroscopy. The apportionment of carbon in different fractions and fluorescence spectra revealed that the OM became less accessible and biodegradable after treatments, especially the composting. This was proven by the decrease in CO_2_ emission from soil incubation. The PLS model successfully predicted the stability of solid digestate, compost, and compost of solid digestate in the soil by using only the characterized variables of non-treated wastes. The results suggested that it would be possible to predict the stability of OM from organic wastes after different treatment procedures. It is helpful to choose the most suitable and economic treatment procedure to stabilize labile organic carbon in wastes and hence minimize CO_2_ emission after the application of treatment products to the soil.

## 1. Introduction

Spreading animal excrements, crop residues, food wastes, and other organic wastes on the farmland to fertilize the soil and enhance soil carbon storage has been an agricultural practice for centuries [1,2,3]. However, the direct spreading of non-treated wastes requires careful control since they mainly consist of labile organic matter (OM), which is not recalcitrant enough to decompose [4]. The decomposition of labile OM in the soil risks releasing excess ammonium and toxic compounds, e.g., phenolic compounds and ethylene, which can hinder plant growth [5,6,7]. The application of immature organic matter can result in inhibited seed germination, root destruction, suppressed plant growth, and a decrease in oxygen concentration and redox potential [8,9]. Moreover, the labile OM leads to a “priming effect” in the soil and multiplies CO_2_ emissions from the soil [10,11]. Only stabilized OM can bring a net accumulation of soil carbon storage in the long term [12].

Anaerobic digestion (AD) and composting are typical biological treatments of organic wastes [13]. The demand for renewable energy has stimulated the boom of AD plants in recent years and brought a huge amount of digestate. The solid part of raw digestate has been used to replace fully or partially chemical fertilizers in many countries [14,15]. Nowadays, there is a growing market of more than 2 000 AD and 3 700 composting plants operational in Europe [16]. However, there is an increasing awareness that the solid digestate needs to be stabilized further via composting before soil application because of its phytotoxicity and poor stability [17,18]. Thus, organic wastes are generally managed through four procedures: (1) direct spreading; (2) AD-spreading (solid digestate); (3) composting-spreading; and (4) AD-composting-spreading. To choose the most environmentally friendly management practice, it would be ideal to predict the stability of OM spread into the soil following these four procedures [19,20,21,22].

Various AD models have been developed, ranging from steady-state to complex dynamic models [23,24,25]. Although a large step forward has been made, a model capable of predicting the properties of digestate is still absent [26,27]. Moreover, it is difficult to couple an AD model to a composting or soil one since their inputs and outputs are different [28]. Dozens of indicators have been proposed in composting and soil fields [29]. However, none of these could appropriately elucidate the stability of OM in the solid digestate [30]. Recently, a new OM characterization protocol consisting of sequential extraction and fluorescence analysis was proposed [31,32]. The apportionment of carbon (AC, percentage of total carbon in the sample) in different fractions out of sequential extraction was used to indicate the accessibility of OM to microorganisms, i.e., the OM in a readily extractable fraction is more accessible than that in a poorly extractable fraction [33,34]. The 3D fluorescence spectra of the supernatant were used to indicate the complexity of soluble OM [35]. Using this characterization protocol, previous works predicted successfully the stability of solid digestate [35,36].

This work proposes combining this OM characterization protocol with partial least squares regression (PLS) modeling to predict the stability of OM in treatment products originating from different waste treatment procedures, e.g., non-treated wastes, solid digestate, compost, compost of solid digestate. A total of 31 organic wastes and their treatment products were characterized using the characterization protocol. Three PLS sub-models were developed for anaerobic digestion, composting, and soil, respectively. These sub-models can be coupled according to users’ requirements to predict the non-mineralized carbon (C_nm_) of OM once applied to the soil.

## 2. Materials and Methods

### 2.1. Description of Samples

Non-treated organic wastes (*n* = 31), solid digestate (*n* = 23), and compost of solid digestate and wastes (*n* = 34) were collected from farmlands, waste treatment plants, or experimental pilot plants in France (Table 1). The solid digestate at the industrial scale was directly collected at the output of the liquid/solid phase separation unit on site. The raw digestate of AD at the pilot scale was also pressed and sieved to remove the liquid phase. The compost was sieved at 20 mm to remove impurities and large woody tissues. Only the fine fraction of compost (<20 mm) was collected.

### 2.2. Sequential Chemical Extraction

The sequential extraction consists of four steps with different chemical solutions (extractants) [37,38]. The five fractions were named according to this sequence: Soluble fraction from Particular extractable Organic Matter (SPOM), Readily Extractable Organic Matter (REOM), Slowly Extractable Organic Matter (SEOM), and Poorly Extractable Organic Matter (PEOM). The fraction of Non-Extractable Organic Matter (NEOM) was left in the precipitate after the sequential extraction. The AC in five fractions (SPOM_AC, REOM_AC, SEOM_AC, PEOM_AC, and NEOM_AC) revealed the chemical accessibility of carbon (OM).

The protocol is presented in Table 2. The fresh samples were first dried at 40 °C until their mass losses were constant. The mass loss was considered constant once the mass loss within 24 h was inferior to 0.5% of that of the previous 24 h. The dry samples were then ground to powder (1.0 mm). Around 0.5 g of powder samples in duplicate were weighted and extracted on a shaker (300 rpm). After extraction, the supernatant and precipitate were separated by centrifugation at 18 600 g, 4 °C for 20 min after each agitation. The supernatant was filtered at 0.45 µm (PTFE) and stored at −20 °C for further analyses. In order to verify the reproducibility of the protocol, this sequential extraction was conducted at least in triplicate for each sample.

### 2.3. Chemical Analyses

The total carbon content (TC) of the solid sample, e.g., the initial powder sample, the final precipitate (NEOM), was analyzed by an element analyzer (NA1500, CARLO ERBP INSTRUMENTS). The TC was expressed per gram of dry matter of the initial powder sample (mg·g^−1^ DM). The total organic carbon (TOC) of each supernatant (SPOM, REOM, SEOM, and PEOM) was analyzed by a TOC analyzer (TOC-5050P, SHIMADZU). The TOC of supernatant was also expressed per gram of dry matter of the initial powder sample (mg·g^−1^ DM). The AC (apportionment of carbon) was calculated from the carbon content (TC or TOC) of each fraction divided by that of the initial powder sample. In this way, the OM of wastes, digestates, and composts was characterized by the carbon content of OM (OM_TC, mg·g^−1^ DM) and 5 accessibility variables: SPOM_AC, REOM_AC, SEOM_AC, PEOM_AC, and NEOM_AC.

The concentration of CO_2_ trapped in the NaOH solution during the incubation was analyzed by the same TOC analyzer. All the chemical analyses were also conducted in triplicate. Sequential extraction and chemical analyses could be repeated if the standard deviation among triplicates was larger than 10%. The mean values of chemical analyses were used for further analysis.

### 2.4. Three-Dimensional Fluorescence Analysis

It should be noted that Chen, Westerhoff [39] proposed fluorescence regional integration (FRI), separating the spectra into five zones. Here this protocol made a more detailed division of regions IV and V in FRI into zones III, V, VI, and VIII, which helped to discriminate easily humic acid-like and lipofuscin-like materials (Appendix A). For more detail, please refer to Jimenez, Aemig [31], Aemig, Chéron [40], and Fernández-Domínguez, Patureau [36]. The fluorescence proportions (P_f_) of seven zones (I–VII) were normalized and calculated based on fluorescence intensity and zone volume (Equations (1) and (2)).
(1)$$V_{f}\left(i\right)\left(U.A./mgTOC\cdot L^{-1}\right)=V_{f\_raw}\left(i\right)/TOC_{sample}\times 1/\frac{S\left(i\right)}{\sum_{i=1}^{7}S\left(i\right)}$$
(2)$$P_{f}\left(i\right)\left(\%\right)=\frac{V_{f}\left(i\right)}{\sum_{i=1}^{7}V_{f}\left(i\right)}\times 100$$
where $V_{f}\left(i\right)$ is the normalized volume of a zone i ($U.A./mgO_{2}\cdot L^{-1}$), $V_{f\_raw}$ is the raw fluorescence volume of a zone i ($U.A./mgO_{2}\cdot L^{-1}$), $TOC_{sample}$ is the TOC concentration of the sample ($mg\cdot L^{-1}$), $S\left(i\right)$ is the area of a zone i (nm^2^), and $P_{f}\left(i\right)$ is the fluorescence proportion of a zone i (%).

In this way, the OM of wastes, digestates, and composts was characterized by 28 complexity variables: SPOM_P_f_ (I–VII), REOM_P_f_ (I–VII), SEOM_P_f_ (I–VII), and PEOM_P_f_ (I–VII).

### 2.5. Incubation in the Soil

The non-mineralized carbon (C_nm_) of samples was obtained from soil incubation, which is like other respiration indicators. A low CO_2_ emission during the incubation signified a high C_nm_, and hence high stability of OM in this sample once spread in the soil. The incubated samples included manure, biowaste, mixed wastes, sludge, vegetable residues, and the corresponding digestate and composts (Appendix A). Biowaste was not incubated because this type of waste was not allowed to be directly spread according to French regulations.

The soil of silty loam texture was collected at 0–5 cm depth from our experimental farmland site near Versailles, Yvelines, France (48°50′23” N, 1°56′50” E). It consisted of 18% clay, 73% silt, 8% sand, and 2.3% organic matter. Its pH and C/N ratio were 6.6 and 12, respectively. The collected soil was air-dried, homogenized, and screened at 5.0 mm, then stored at 4 °C until it was used. The particle sizes of samples were different, which also plays an important role in biodegradation. The higher specific surface of smaller particles facilitates biodegradation. In order to avoid this effect, like sequential extraction, all the samples were also dried and ground to 1.0 mm prior to the incubation.

Before incubation, the 1.0 mm shredded samples were mixed with dry soil at a ratio equivalent to 4 g C kg^−1^ dry soil. A suitable quantity of water was then added to make the humidity of the mixture equivalent to 28% (*w*/*w*). The moist mixture was thereafter loaded into 3 L glass jar and incubated at 28 °C in the dark in growth chambers for 175 days. Soil moisture was maintained through the incubation by weighing the soil at weekly intervals and adding deionized water when necessary. The water-filled pore space (WFPS) of the soil pores was held at around 65%. The WFPS was calculated according to references [41,42]. A control treatment without any OM amendment was also included. Each sample was incubated in four replicates.

The CO_2_ emission was measured from the OM-amended soil during the incubation using a CO_2_ trap (50 mL of 1 M NaOH) in the jar. The mean captured CO_2_ was measured in four replicates at 1, 3, 7, 14, 21, 28, 49, 70, 91, 112, 133, 154, and 175 days after the beginning by replacing the CO_2_ traps at those dates. The net CO_2_ production was calculated from the difference of mean captured CO_2_ values between the OM-amended soil and the control, under the assumption that the mineralization of native soil organic C was not significantly modified by the addition of OM (no priming effect) or that the priming effect was of the same order of magnitude in all tested substrates. The net CO_2_ production was then transformed and expressed per gram of dry matter of the sample. The C_nm_ was calculated by subtracting the amount of C released as net CO_2_ production from the total C content of the sample (Appendix A).

### 2.6. Statistical Analysis and Models Building

The statistical analysis and PLS regression were performed by using the software SIMCA-Plus 14.1 (MKS Umetrics). The principal components analysis (PCA) was fitted by choosing the smallest number of principal components that were required in order to explain a large amount of the variation in the data [43,44]. This was performed by checking the ordinal number of principal components at which the proportion of variance explained by each subsequent principal component dropped off.

Three sub-models were developed in the PLS approach for anaerobic digestion, composting, and soil (Figure 1), respectively. The PLS is a dimension reduction method and a supervised alternative to principal components regression (PCR), which attempts to find directions (principal components) that help explain both the response and the predictors. The input of the digestion and composting sub-models was 34 featuring variables of OM before treatment: the carbon content of OM (OM_TC, mg·g^−1^ DM), 5 accessibility variables (SPOM_AC, REOM_AC, SEOM_AC, PEOM_AC, and NEOM_AC), and 28 complexity variables (SPOM_P_f_ (I–VII), REOM_P_f_ (I–VII), SEOM_P_f_ (I–VII), and PEOM_P_f_ (I–VII)). The output was 34 featuring variables of OM in digestate and compost. The input of the soil sub-model was also 34 variables, while its output was the C_nm_ of OM in the soil. There were also some parameters to evaluate the performance of sub-models. R^2^X (cum) was the percent variation in the X matrix (input) explained by the sub-model, while R^2^Y (cum) was the percent variation in the Y matrix (output) explained. R^2^X (cum) and R^2^Y (cum) were measures of fit, i.e., how well the sub-model fit the data. Q^2^ (cum) was the percent variation predicted by the sub-model according to cross-validation. Q^2^ (cum) was a measure of predictivity, i.e., how well the sub-model predicts new data. All the data of this work can be downloaded in Appendix A.

## 3. Results and Discussion

### 3.1. Grouping of Non-Treated Organic Wastes

31 organic wastes (input, Table 1) were clustered using hierarchical cluster analysis (HCA) on 34 variables (Figure 2). The wastes in Group 1 were the raw sludge (Sludge1–3), while Group 2 consisted of the mixture of digested sludge (Mix8-D) and the mixture of manure, food waste, and turf (Mix10). The cow/beef manures mixed with straw/hay (Manure2–5), straws, and corn stalks made up Group 3. Group 4 included green waste, poultry manure mixed with straw (Manure1), and mixtures of raw sludge with green waste (Mix1–4). At last, all the wastes containing a fine organic fraction of household waste (Biowaste1–5), Manure6, Mix5, Mix6, Mix7, and Mix9-D formed Group 5. The Manure6 sample was different from other manures since it was a supernatant of centrifuged pig manure mixed with horse fodder. Different from Mix1–4, Mix5–7, and Mix9-D were mixtures of raw sludge or sludge digestate with grass and tree bark. Despite the heterogeneity of various organic wastes, the result of HCA suggested that the 34 variables could help to distinguish these wastes in terms of their origins and compositions.

### 3.2. Influence of Biological Treatment on Organic Matter of Wastes

The 34 variables of OM in all the samples, including non-treated wastes (*n* = 31), digestate (*n* = 23), and composts (*n* = 34), were analyzed using PCA to visualize the data and explore the potential differences among wastes, digestate, and composts. The 34 variables were reduced to five principal components (R^2^ = 0.777, Q^2^ = 0.553). The first two components explained 56% of the variance in the data (Figure 3A. The red points represented the organic wastes before treatment. The blue points and green points were the digestate and compost, respectively. There was a trend of aggregation from the right to the upper left. A large part of organic wastes (red points) was on the first and second quadrants of the ellipse. The digestate (blue points) dispersed in the middle of the ellipse. The composts were (green points) assembled on the fourth quadrant.

To improve the visibility, Figure 3A was zoomed in on and divided into four sub-figures corresponding to five groups of organic wastes classified by HCA (Figure 3B−E). The Sludge1 and Sludge2 samples were closely located on the right boundary of the ellipse because they were sampled from the same WWTP at different dates (Figure 3B). The Sludge3 contained only the secondary sludge, which made it far away from other wastes. This suggested that Sludge3 was an outlier that should be removed in further modeling. Sludge1 and 2 moved to the left after digestion, while the composting made Sludge3 shift to the top left of the ellipse. The Mix8-D was a mixture of sludge digestate. Its compost also moved to the top left. The composition of Mix10 was complicated. It consisted of cow manure, grass, fruits, vegetables, and dietary fat. The addition of dietary fat made this mixture very different from other “low-fat” wastes. It was digested under three reactors: thermophilic condition (Mix10-D1 and -D2) and mesophilic (Mix10-D3). It seems the thermophilic digestion changed the characteristics of the mixture since Mix10-D1 and -D2 were far away from Mix10, while the mesophilic digestate (Mix10-D3) was near to Mix10. The wastes in Group 3 were found around the origin (Figure 3C). Like Group 1, their digestate moved to the upper left. Except for Stalk1, the digestate was always located on the upper left of undigested waste. The Stalk2-98d sample was not a digestate, but the corn stalk spread and was left on the top of the soil for 98 days. It seems soil spreading significantly changed the characteristics of the corn stalks.

The right-to-upper-left shift became more obvious in Figure 3D,E, except for the Manure6 and Mix7-C1 samples. The mature composts were far away from their original wastes and assembled on the top left of the ellipse, while the digestate was scattered between wastes and composts. The green points with positive scores in the first component were, in fact, wastes composted for only 7–14 days. In other words, they were immature. The Manure6-D was probably not stabilized due to the poor stirring of the digester, which made it like other non-treated wastes [45]. Mix7-C1–3 were composts sampled at 7, 28, and 70 days, respectively (Table 1). Only Mix7-C1 was located on the right side of the original waste. It implied biowaste did not always become more stabilized throughout the whole composting period. Organic matter in biowaste could become more hydrolyzed, in other words, more unstable, at the beginning (7 days) of the composting due to strong microbial activities. However, other samples did not show this tendency because they were all collected after 13 days of composting. Since there were not enough similar samples, Mix7-C1 was still considered an outlier and removed in modeling. This exploratory analysis of the PCA score scatter plot indicated, for a given waste, that there was a right-to-upper-left shift among waste, digestate, and compost.

The loadings of the first two components are displayed in Figure 4 to help our understanding of this right-to-upper-left shift. The loadings of five components are also provided in the data table in Appendix A. The first principal component loading vector placed almost all its weight on P_f_ of zones I–III (SPOM_P_f_ (I–III), REOM_P_f_ (I–III), SEOM_P_f_ (I–III), and PEOM_P_f_ (I–III)), which indicated “simple” protein-like materials in four extractable fractions. The variables indicating “complex” organic matter (SPOM_P_f_ (IV–VII), REOM_P_f_ (IV–VII), SEOM_P_f_ (IV–VII), and PEOM_P_f_ (IV–VII)) were in the opposite direction of “simple” materials. This right-to-upper-left shift signified, in fact, an increase in the complexity of molecules. Moreover, the apportionment of carbon in easily extractable fractions (SPOM_AC, REOM_AC, SEOM_AC) contributed positively to the first principal component and negatively to the second principal component, while that of poorly extractable organic matter (PEOM_AC) and non-extractable organic matter (NEOM_AC) were in opposition. This right-to-upper-left shift also suggested a decrease in accessibility. The OM in wastes became less accessible and more complex after AD or composting. Almost all the compost is assembled in a small area to the left of the digestate. Composting was capable of further stabilizing the digestate and producing similar composts regardless of the origins of wastes or digestate. This result confirmed our hypothesis. The 34 variables revealed the evolution of the stability of OM in wastes throughout treatments.

The Van Soest method is one of the most widely used OM fractionation protocols [46]. According to the Van Soest method, the OM can be separated into neutral detergent fiber (NDF), acid detergent fiber (ADF), and strong acid detergent fiber (SADF). Various protocols modified from the Van Soest method and other similar extraction methods have been applied to OM characterization [47,48]. In combination with these extraction methods, various kinds of models have been developed to predict the biodegradation of OM in the soil, digestion, composting, etc. [49,50,51]. According to these models, e.g., CANTIS, NCSOIL, and COP-Compost, different mineralization rate values were allocated to these extracted fractions [50,52,53]. However, the extraction sequence could only reflect the accessibility of OM to microorganisms [54]. The composition of each fraction might evolve. This made the allocated value of the same fraction different from one substrate to another in different studies.

This work suggested both accessibility and complexity should be considered while evaluating the biodegradation of OM. The HCA analysis indicated this sequential extraction and fluorescence analysis protocol could distinguish the origins and compositions of organic wastes. The first two components of PCA implied the “complexity” of accessible OM should indeed be considered. Although the physical meanings of principal components were unclear, which was the main drawback of PCA, the loading vectors of the first two components clearly revealed the changes in the molecular structures of OM through treatments. A regression approach could therefore be performed using the principal component score vectors as features to predict the stability of OM with much less noisy results [55].

### 3.3. Development of PLS Sub-Models

Three PLS sub-models were built for AD, composting, and soil, respectively (Figure 1). The data of Manure1, Biowaste2, and their digestate, composts, and composts of digestate were excluded from building the sub-models but reserved for further validation.

The inputs of the AD sub-model included 34 variables of OM in wastes and the hydraulic retention time (days). The output of the AD sub-model was 34 variables of OM in the digestate. The hydraulic retention time was found to be more relevant in predicting the 34 output variables than other process parameters, e.g., the mesophilic/thermophilic condition and the dry/humid process (Table 1). A regression approach could therefore be performed using the principal component score vectors as features to predict the stability of OM with much less noisy results. The R^2^X (cum) and R^2^Y (cum) of the AD sub-model up to the third component were 0.898 and 0.678, respectively, which indicated a good fit for the sub-model. The Q^2^Y (cum) of the AD sub-model was 0.524, which was superior to 0.5 and indicated good predictivity [56].

In the same way, the inputs of the composting sub-model included 34 variables of OM in the waste/digestate and the duration of composting (days). The output of the composting sub-model was 34 variables of OM in the compost. The outliers (Sludge3 and Mix7-C1) identified in Section 3.2 were also removed. The R^2^X (cum), R^2^Y (cum), and Q^2^Y (cum) of the composting sub-model were 0.935, 0.812, and 0.534, respectively. To further check the quality of the sub-models, Figure 5A, B show the cumulated R^2^ and Q^2^ values for each variable in the above two sub-models. Here, R^2^VY (cum) indicated how well the variation in a variable was explained, while Q^2^VY (cum) indicated how well a variable could be predicted. For most of the variables, their R^2^VY (cum) was close to 0.8. The Q^2^VY (cum) was also above 0.5. This suggested a good fit and predictivity for most of the variables in the sub-models.

The soil sub-model used the 34 variables of OM to predict their C_nm_ after spreading in soil. The R^2^X(cum), R^2^Y(cum), and Q^2^Y(cum) of the soil sub-model were 0.605, 0.947, and 0.753, respectively. Figure 5C displays the observed versus predicted C_nm_. Almost all the points fell close to this 45-degree line, which indicated a good predictivity of the sub-model. The root-mean-square error of estimation (RMSEE), which indicated the fit of the observations to the sub-model, was only 3.803. The root-mean-square error of co-variance (RMSEcv) is analogous to RMSEE but estimated using cross-validation. The RMSEcv of the soil sub-model was 8.463.

### 3.4. Coupling of Three PLS Sub-Models

Three sub-models were coupled to predict the C_nm_ of OM in the soil, e.g., predicting the C_nm_ of Manure1-D-C in the soil using the 34 variables of only Manure1. The performance of prediction was validated using the data of Manure1, Biowaste2, and their treatment products. The observed vs. predicted values of 34 variables are given in the data table in Appendix A. The observation vs. prediction result of C_nm_ is presented in Figure 6 (blue bars vs. orange bars). The C_nm_ of Biowaste2 was absent since it was not incubated. The gap between the observation prediction of Manure1 was larger than that of others. The observed C_nm_ of Manure1 was only 50.8%, but the model returned 64.4%. Taking the diversity of different types of non-treated wastes into consideration, the data was probably insufficient to assure the precise prediction of direct soil-spreading wastes. However, the model obtained satisfactory results in the prediction of C_nm_ for digestate and compost. The model predicted the C_nm_ of digestate of Biowaste2 was only 71.9%. The Biowaste2-D was indeed not sufficiently stabilized. The model predicted the digestate was further stabilized after 77 days of composting. The C_nm_ of Biowaste2-D-C2 attained 84.8%, which was superior to that of Biowaste2-D-C1 (82.6%) with 50 days of composting.

To further verify the need for taking both accessibility and complexity into consideration, we built another two sets of three sub-models using the same PLS approach to predict the C_nm_ (Figure 6). The first set used only the carbon content of OM plus accessibility variables (6 variables, grey bars), while the second one used only the carbon content of OM plus complexity variables (29 variables, yellow bars). It was clear that using only accessibility variables trended to overestimate the C_nm_ in most cases. In contrast, using only complexity variables could underestimate the C_nm_ while coupling the sub-models. This comparison suggests both accessibility and complexity variables were necessary to predict the stability of OM in the soil.

Various anaerobic digestion, composting, and soil models have been developed, ranging from steady-state to statistical learning and dynamic models [23,24,57]. However, there are few anaerobic digestion models focusing on the biodegradability of OM in digestate [58,59]. There are much more models devoted to the stability of OM in compost and soil. However, various indicators have been proposed to evaluate the stability of OM, e.g., EC (electric conductivity), C/N, germination index, humification index, nitrification index, biological denitrification potential, and Shannon index [60,61,62,63]. These indicators are difficult to be unified. Coupling an AD model to a composting or a soil one to predict the stability of OM is extremely difficult since their inputs and outputs are different [28].

This sequential extraction plus fluorescence analysis protocol has been built and improved by our previous studies [31,38,64]. In combination with the PLS regression modeling approach, its performance on the prediction of OM biodegradability has been proved by works on digestate [33,36] and compost of digestate [34,37]. This work is the first time we tried to build three sub-models and couple them together to predict the stability of OM in the soil. The result of coupling three sub-models suggested that for a given waste, we could use the sequential extraction and fluorescence spectroscopy characteristics to predict the stability of its OM in the soil originating from different treatment procedures. Moreover, this modeling approach has other advantages: (1) flexibility, as the sub-models can be easily coupled according to a specific procedure, and (2) the possibility of being improved, as the performance of the model can be improved in pace with the accumulation of the database.

## 4. Conclusions

Organic wastes, digestate, and composts collected from different waste treatment sectors were characterized by using an organic matter characterization protocol. The 34 featuring variables, which included the carbon content, the apportionment of carbon in five fractions, and the fluorescence proportions of seven spectra zones in four soluble fractions, revealed the progressive increase in the stability of organic matter from non-treated wastes to digestate and further, to compost. Three PLS regression sub-models were built for anaerobic digestion, composting, and soil, respectively. Inputting the 34 featuring variables of non-treated wastes into the coupled sub-models could successfully predict the stability of digestate, composts, and composts of digestate in the soil. This modeling approach would help us in choosing the most environmentally friendly treatment procedure according to the stability of organic matter in the wastes, e.g., mitigate greenhouse gas emissions from soil application, enhance the stability of soil carbon storage, and acquire more renewable energy from easily biodegradable wastes.

## Figures and Tables

**Figure 1 ijerph-20-02151-f001:**
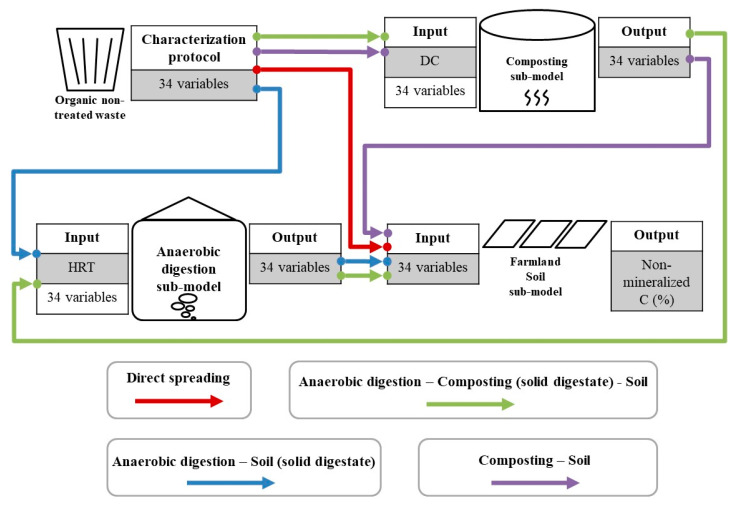
Schema of the concept of modeling. AD−anaerobic digestion; HRT−hydraulic retention time (days); DC−duration of composting (days).

**Figure 2 ijerph-20-02151-f002:**
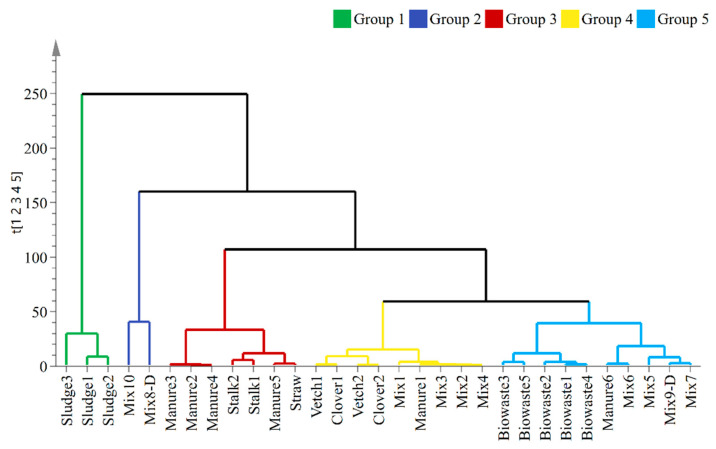
Hierarchical cluster analysis (HCA) of organic wastes.

**Figure 3 ijerph-20-02151-f003:**
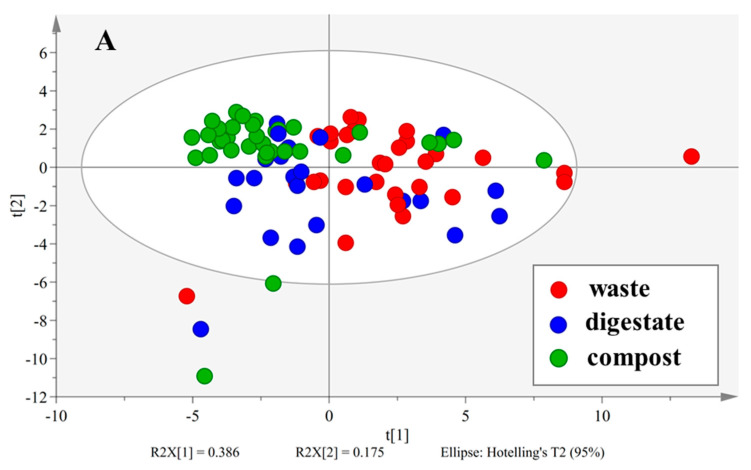
PCA score scatter plot of first two principal components on 34 variables of all the samples (red points−organic waste; blue points−digestate; green points−compost). (**A**) Score scatter plot of all the samples; (**B**) Groups 1 and 2; (**C**) Group 3; (**D**) Group 4; (**E**) Group 5.

**Figure 4 ijerph-20-02151-f004:**
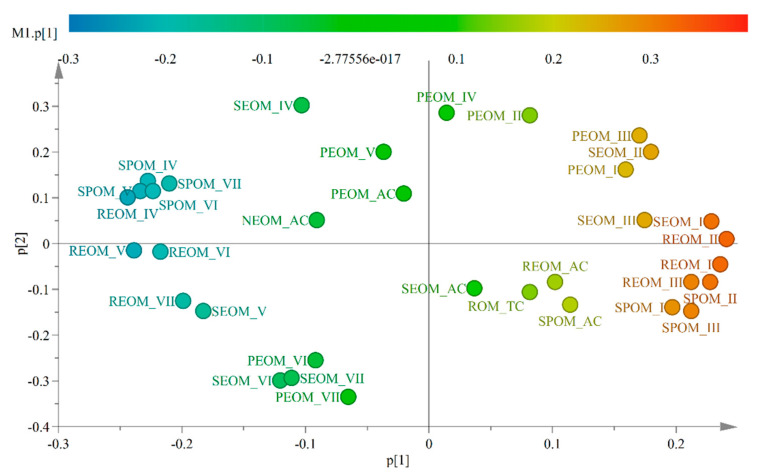
Loading scatter plot for the first two principal components. (The color scale indicates the contribution of 34 variables on the first component. SPOM−soluble fraction from particular extractable organic matter; REOM−readily extractable organic matter; SEOM−slowly extractable organic matter; PEOM: poorly extractable organic matter; AC−apportionment of carbon in corresponding fraction; I−VII−fluorescence proportions of seven zones in the corresponding fraction, the mark “P_f_” was removed to reduce the length of labels).

**Figure 5 ijerph-20-02151-f005:**
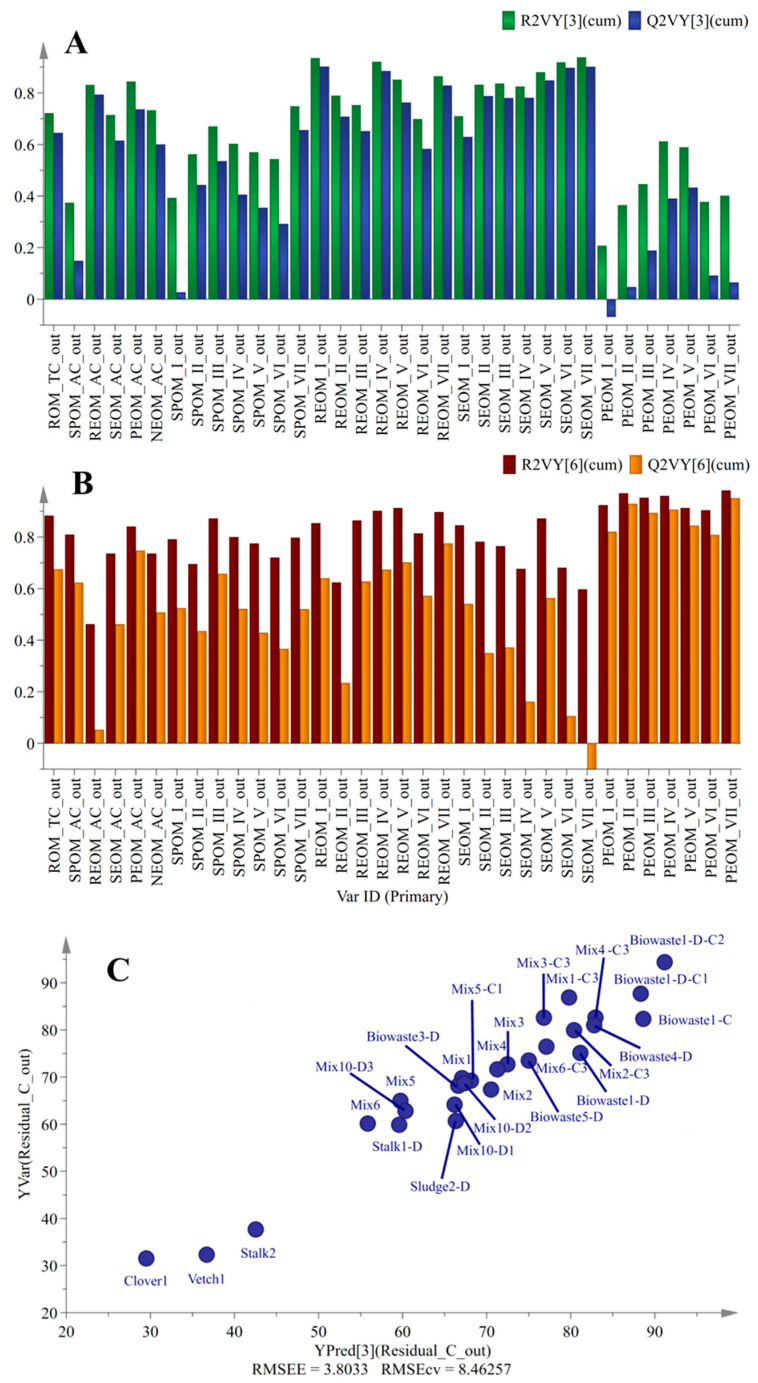
Performance of partial least squares (PLS) regression model. (**A**,**B**) R^2^Y(cum) and Q^2^Y(cum) of each variable for anaerobic digestion and composting sub-models. (**C**) Observed versus predicted C_nm_ of soil sub-model. Out−variables of solid digestate, compost, or compost of solid digestate after anaerobic digestion or composting; I−VII−the fluorescence proportions (P_f_) of seven zones (I−VII) in 3D fluorescence spectra.

**Figure 6 ijerph-20-02151-f006:**
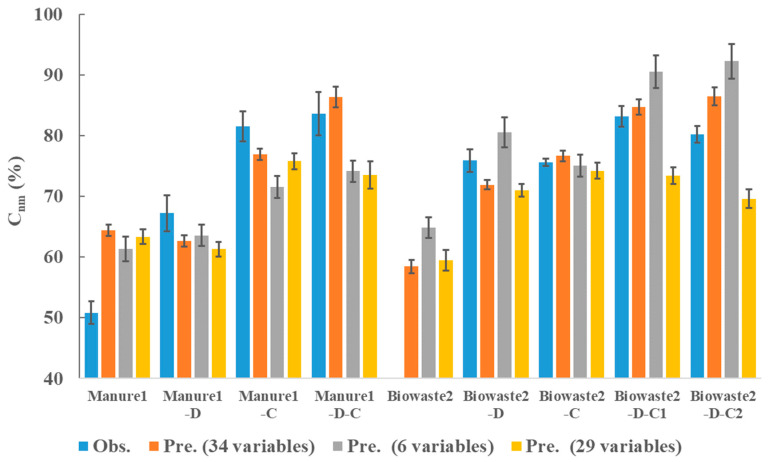
The observed non-mineralized carbon (C_nm_, %) values vs. predicted values. The blue, orange, grey, and yellow bars represent observed C_nm_ value, predicted C_nm_ values using both accessibility and complexity variables (*n* = 34), using only the carbon content of OM plus accessibility variables (*n* = 6), and using only the carbon content of OM plus complexity variables (*n* = 29), respectively. C_nm_—non-mineralized C (%) of OM in the soil; D—digestate; C—compost.

**Table 1 ijerph-20-02151-t001:** Composition of organic wastes and treatment products.

Sample		Description			Sample (Solid Digestate or Compost, Output)
*n*	(Non-Treated Waste, Input)	Treatment	Scale	Phase Separation	
1	Poultry manure mixed with straw (Manure1)	AD ^1^: 41 °C, 70 days ^2^	Industry	Solid phase	Solid digestate of Manure1 (Manure1-D) ^4^
		Composting: 56 days ^3^	Pilot	Fine fraction (sieved at 10 mm)	Compost of Manure1 (Manure1-C) ^5^
2	Cow manure mixed with straw (Manure2)	Dry AD: 35 °C, 56 days	Pilot	Solid phase	Solid digestate of Manure2 (Manure2-D)
3	Beef manure mixed with straw (Manure3)	Dry AD: 35 °C, 56 days	Pilot	Solid phase	Solid digestate of Manure3 (Manure3-D)
4	Cow and beef manure mixed with straw (Manure4)	Dry AD: 35 °C, 29 days	Industry	Solid phase	Solid digestate of Manure4 (Manure4-D1)
		Dry AD: 35 °C, 56 days			Solid digestate of Manure4 (Manure4-D2)
5	Cow and beef manure mixed with straw and hay (Manure5)	Dry AD: 35 °C, 56 days	Industry	Solid phase	Solid digestate of Manure5 (Manure5-D)
6	Centrifuged pig manure mixed with horse fodder (Manure6)	AD: 35 °C, 70 days	Pilot	Solid phase	Solid digestate of Manure6 (Manure6-D)
7	Fine organic fraction of household waste (Biowaste1)	Dry AD: 55 °C, 28 days	Industry	Solid phase	Solid digestate of Biowaste1 (Biowaste1-D)
		Composting: 50 days	Pilot	Fine fraction (sieved at 10 mm)	Compost of Biowaste1 (Biowaste1-C)
8	Fine organic fraction of household waste mixed with green wastes (Biowaste2)	AD: 55 °C, 21 days	Industry	Solid phase	Solid digestate of Biowaste2 (Biowaste2-D)
		Composting: 50 days	Pilot	Fine fraction (sieved at 10 mm)	Compost of Biowaste2 (Biowaste2-C)
9	Fine organic fraction of household waste mixed with green wastes (Biowaste3)	Dry AD: 55 °C, 21 days	Industry	Solid phase	Solid digestate of Biowaste3 (Biowaste3-D)
10	Fine organic fraction of household waste mixed with green wastes (Biowaste4)	Dry AD: 37 °C, 28 days	Industry	Solid phase	Solid digestate of Biowaste4 (Biowaste4-D)
11	Fine organic fraction of household waste mixed with green wastes and papers (Biowaste5)	Dry AD: 53 °C, 20 days	Industry	Solid phase	Solid digestate of Biowaste5 (Biowaste5-D)
12	Primary sludge mixed with secondary sludge (Sludge1)	AD: 35 °C, 20 days	Pilot	Solid phase	Solid digestate of Sludge1 (Sludge1-D1)
		AD: 35 °C, 20 days			Solid digestate of Sludge1 (Sludge1-D2)
		AD: 35 °C, 20 days			Solid digestate of Sludge1 (Sludge1-D3)
		AD: 55 °C, 15 days	Industry	Solid phase	Solid digestate of Sludge1 (Sludge1-D4)
13	Primary sludge mixed with secondary sludge (Sludge2)	AD: 37 °C, 20 days	Industry	Solid phase	Solid digestate of Sludge2 (Sludge2-D)
14	Waste activated sludge (Sludge3)	Composting: 60 days	Industry	Fine fraction (sieved at 10 mm)	Compost of Sludge3 (Sludge3-C)
15	Wheat straw (Straw)	Dry AD: 35 °C, 56 days	Pilot	Solid phase	Solid digestate of Straw (Straw-D)
16	Corn stalks (Stalk1)	Dry AD: 50 °C, 50 days	Industry	Solid phase	Solid digestate of Stalk1 (Stalk1-D)
17	Mixture of sewage sludge, green waste, branches, and grass clippings (Mix1)	Composting: 14 days	Pilot	Fine fraction (sieved at 10 mm)	Compost of Mix1 (Mix1-C1)
		Composting: 42 days			Compost of Mix1(Mix1-C2)
		Composting: 84 days			Compost of Mix1(Mix1-C3)
18	Mixture of sewage sludge, green waste, branches, and grass clippings (Mix2)	Composting: 14 days	Pilot	Fine fraction (sieved at 10 mm)	Compost of Mix2 (Mix2-C1)
		Composting: 42 days			Compost of Mix2 (Mix2-C2)
		Composting: 84 days			Compost of Mix2 (Mix2-C3)
19	Mixture of sewage sludge, green waste, branches, grass clippings and pallet (Mix3)	Composting: 14 days	Pilot	Fine fraction (sieved at 10 mm)	Compost of Mix3 (Mix3-C1)
		Composting: 42 days			Compost of Mix3 (Mix3-C2)
		Composting: 84 days			Compost of Mix3 (Mix3-C3)
20	Mixture of sewage sludge, green waste, branches, grass clippings and corn stalks (Mix4)	Composting: 14 days	Pilot	Fine fraction (sieved at 10 mm)	Compost of Mix4 (Mix4-C1)
		Composting: 42 days			Compost of Mix4 (Mix4-C2)
		Composting: 84 days			Compost of Mix4 (Mix4-C3)
21	Mixture of sewage sludge and pallet (Mix5)	Composting: 14 days	Pilot	Fine fraction (sieved at 10 mm)	Compost of Mix5 (Mix5-C1)
		Composting: 42 days			Compost of Mix5 (Mix5-C2)
		Composting: 84 days			Compost of Mix5 (Mix5-C3)
22	Mixture of sewage sludge, branches, and grass clippings (Mix6)	Composting: 14 days	Pilot	Fine fraction (sieved at 10 mm)	Compost of Mix6 (Mix6-C1)
		Composting: 42 days			Compost of Mix6 (Mix6-C2)
		Composting: 84 days			Compost of Mix6 (Mix6-C3)
23	Mixture of sewage sludge, grass, and tree bark (Mix7)	Composting: 7 days	Pilot	Fine fraction (sieved at 10 mm)	Compost of Mix7 (Mix7-C1)
		Composting: 28 days			Compost of Mix7 (Mix7-C2)
		Composting: 70 days			Compost of Mix7 (Mix7-C3)
24	Mixture of three sludge digestate (Mix8-D)	Composting: 60 days	Industry	Fine fraction (sieved at 20 mm)	Compost of Mix8-D (Mix8-D-C)
25	Mixture of sludge digestate, grass and tree bark (Mix9-D)	Composting: 13 days	Pilot	Fine fraction (sieved at 10 mm)	Compost of Mix9-D (Mix9-D-C1)
		Composting: 30 days			Compost of Mix9-D (Mix9-D-C2)
		Composting: 70 days			Compost of Mix9-D (Mix9-D-C3)
26	Mixture of manure, turf, fruit, vegetable, and dietary fat (Mix10)	AD: 35 °C, 75 days	Industry	Solid phase	Solid digestate of Mix10 (Mix10-D1)
		AD: 35 °C, 75 days			Solid digestate of Mix10 (Mix10-D2)
		AD: 55 °C, 75 days			Solid digestate of Mix10 (Mix10-D3)
27	Corn stalks (Stalk2)	Shredded to <10 cm then spread on the top of the soil, 98 days	Farmland	Fine fraction (sieved at 20 mm)	Residue of Stalk2 (Stalk2-98d)
28	Clover (*Trifolium sp.*) reaped in December (Clover1)	―	―	―	―
29	Clover reaped in Mars (Clover2)				
30	Vetch (*Vicia sativa*) reaped in December (Vetch1)				
31	Vetch reaped in Mars (Vetch2)				
**Sample**		**Description**			**Sample**
*n*	**(Solid digestate, Input)**	**Treatment**	**Scale**	**Phase separation**	**(Compost, Output)**
1	Manure1-D	Composting: 56 days	Pilot	Fine fraction (sieved at 10 mm)	Compost of Manure1-D (Manure1-D-C)
2	Biowaste1-D	Composting: 28 days	Industry	Fine fraction (sieved at 20 mm)	Compost of Biowaste1-D (Biowaste1-D-C1)
		Composting: 50 days	Pilot	Fine fraction (sieved at 10 mm)	Compost of Biowaste1-D (Biowaste1-D-C2)
3	Biowaste2-D	Composting: 77 days	Industry	Fine fraction (sieved at 20 mm)	Compost of Biowaste2-D (Biowaste2-D-C1)
		Composting: 50 days	Pilot	Fine fraction (sieved at 10 mm)	Compost of Biowaste2-D (Biowaste2-D-C2)

^1^ AD−anaerobic digestion. ^2^ Hydraulic retention time (HRT) of anaerobic digestion. ^3^ Duration of composting. ^4^ The mark “-D” implies it was a solid digestate. ^5^ The mark “-C” indicates it was compost.

**Table 2 ijerph-20-02151-t002:** Protocol of sequential extraction.

Extracted Fraction	Extractant	Volume of Extractant	Temperature	Agitation	Extraction Duration and Repetition
SPOM ^1^	CaCl_2_ (0.01 M)	30 mL	30 °C	300 rpm. horizontal	15 min × 4
REOM ^2^	NaCl/NaOH (0.01 M)				15 min × 4
Pre-treatment	HCl				1 h × 1
	Ultra-pure water (pH adjusted to 7.0)				5 min × 1
SEOM ^3^	NaOH (0.1 M)				4 h × 4
PEOM ^4^	H_2_SO_4_ (72%)				3 h × 2

^1^ SPOM—soluble fraction from particular extractable organic matter. ^2^ REOM—readily extractable organic matter. ^3^ SEOM—slowly extractable organic matter. ^4^ PEOM—poorly extractable organic matter.

## Data Availability

The data of this research can be found in the Appendix A.

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
