# Peer review of "Predicting the Stability of Organic Matter Originating from Different Waste Treatment Procedures"

_ijerph, 2023, doi:10.3390/ijerph20032151_

Round 1
Reviewer 1 Report
This manuscript reports a modelling approach integrating an OM characterization protocol into partial least squares (PLS) regression. The results showed that OM became less accessible and biodegradable for microbes after treatments. The PLS model predicted successfully the non-mineralized carbon of OM in the soil. The present data are interesting and the topic of this manuscript fits certainly under the scope of Int. J. Environ. Res. Public Health. The manuscript is generally well written and the outcomes are of high general scientific interest.
Please see below for other suggestions to improve the MS:
1. Lines 35-37, except for releasing excess ammonium and toxic compounds, the application of immature organic matter can also result in inhibited seed germination, root destruction, suppressed plant growth, and a decrease in oxygen concentration and redox potential.
2. Line 38, this phenomenon is well known as “priming effects” in soil science. It is better for the authors to mention it.
3. Line 65, the 3D fluorescence spectra can only assess the properties of soluble/labile OM rather than the bulk OM. So it is better to change “the complexity of OM” to “the complexity of soluble OM”.
4. Lines 128-129, the authors separated fluorescence spectra into seven zones. To my knowledge, Chen et al. (2003) proposed fluorescence regional integration (FRI) that separated the spectra into five zones is more popular. It is suggested the authors to discuss why they did not use the FRI analysis to explore the fluorescence spectra.
5. Figure 3. Some text in this figure are too small to be identified.
Reviewer 2 Report
I suggest to accept the proposed manuscript entitled: " Predicting the stability of organic matter originating from different waste treatment procedures" after minor revision => corrections to text editing
Reviewer 3 Report
The revised paper develops three models to predict the stability of organic matter in non-treated wastes, solid digestate, compost and compost of solid digestate. Overall in general terms the manuscript is well-written, it might be further improved if the following can be addressed:
1. The introduction should be rewritten to provide sufficient background in the topics and justify the novelty of this research. The objective of the paper should be clarified and justified: the use of a combination of sequential extraction and fluorescence spectroscopy to predict the stability of non-treated wastes or treatment products in the soil. In this sense it is necessary a further background in this topic including similar studies predicting the stability of organic matter in non-treated and treated waste as well as the use of fluorescence spectroscopy.
2. Lines 55 to 71 include the description of methods used so they should be included in Materials and Methods section. Introduction should finish with the objective of the paper, after a properly motivation.
3. A total of 88 samples are included in section 2.1 (31 non-treated organic wastes, 23 solid digestate and 34 compost of solid digestate and wastes) however Table 1 does not include this number of samples. Please, clarify this.
4. Samples in Table 1 should be included in three different section: non-treated organic wastes, solid digestate and compost of solid digestate and wastes.
5. In general terms sections 2.2 to 2.6 are hardunderstanding. They are too dense. Please revise them and try to make them more understanding. For example in the case of Table 2 it could be completed including the chemical analysis of each fraction.
6. Section 2.6 explains that the input of digestion and composting sub-models was 34 featuring variables of OM before treatment. Please, describes these variables.
7. Section 3.1. should be included in materials and methods section.
8. Figure 2 includes 31 organic wastes from Table 1 but not all the organic wastes in Table 1 has been considered. Please, clarify this.
9. Principal components method has been used to identified the 5 most relevant variables, but the most relevant components have not been included. Besides, the first two components explained 56% of the variance in the data has not been also included in the text. I think that it is necessary to revise results for better presentation of most relevant results as well as a better discussion to justify them
10. In line 319 to 321 the following result has been included: "a regression approach could therefore be performed using the principal component score vectors as features to predict the stability of OM with much less noisy results". Include this.
11. Figure 6 shows the observed non-mineralized carbon values vs predicted values including different number of variables. In general terms the observed value is very different to predicted values.
12. Conclusions should be rewritten to include the most relevant variables affecting the prediction models as well as the most important characteristics of these models. Besides, it should include the novelty of the research.
13. Abstract does not describe the most important results and novelty of the research. Please, rewrite it.
Reviewer 4 Report
This is a very interesting review article. Due to this I agree with the importance of this manuscript.
Regarding the possibilities to improve it, I would like to comment the following if it could help to the authors:
1. To highlight in the abstract the goal of the study, its value and the benefits of the results obtained.
2. To introduce more explications for each graphic presented in the results section.
3. To maintain a similar form of the presentation of the figures shown, when it is possible to look more similar configuration.
4. To show discussions of the results with the references shown.
5. To introduce references of this year 2022.
Thanks so much and good job. I like this very much.
